# Abnormal developmental trajectory and vulnerability to cardiac arrhythmias in tetralogy of Fallot with DiGeorge syndrome

Chun-Ho Chan[1], Yin-Yu Lam[1], Nicodemus Wong[1], Lin Geng[1], Jilin Zhang[2], Virpi Ahola[2], Aman Zare[2], Ronald Adolphus Li[2,3], Fredrik Lanner [4,5,6], Wendy Keung[3] & Yiu-Fai Cheung [1,2,3 ✉]

Tetralogy of Fallot (TOF) is the most common cyanotic congenital heart disease. Ventricular dysfunction and cardiac arrhythmias are well-documented complications in patients with repaired TOF. Whether intrinsic abnormalities exist in TOF cardiomyocytes is unknown. We establish human induced pluripotent stem cells (hiPSCs) from TOF patients with and without DiGeorge (DG) syndrome, the latter being the most commonly associated syndromal association of TOF. TOF-DG hiPSC-derived cardiomyocytes (hiPSC-CMs) show impaired ventricular specification, downregulated cardiac gene expression and upregulated neural gene expression. Transcriptomic profiling of the in vitro cardiac progenitors reveals early bifurcation, as marked by ectopic *RGS13* expression, in the trajectory of TOF-DG-hiPSC cardiac differentiation. Functional assessments further reveal increased arrhythmogenicity in TOF-DG-hiPSC-CMs. These findings are found only in the TOF-DG but not TOF-with no DG (ND) patient-derived hiPSC-CMs and cardiac progenitors (CPs), which have implications on the worse clinical outcomes of TOF-DG patients.

[1] Department of Paediatrics and Adolescent Medicine, Li Ka Shing Faculty of Medicine, The University of Hong Kong, Hong Kong, China. [2] Ming Wai Lau Centre for Reparative Medicine, Hong Kong node, Karolinska Institutet, Units 608-613 Building 15 Science Park, Hong Kong, China. [3] Dr. Li Dak-Sum Research Centre, The University of Hong Kong - Karolinska Institutet Collaboration in Regenerative Medicine, The University of Hong Kong, Hong Kong, China. [4] Ming Wai Lau Centre for Reparative Medicine, Stockholm node, Karolinska Institutet, Solnavagen 9, 17165 Stockholm, Sweden. [5] Department of Clinical Sciences, Intervention and Technology, Karolinska Institutet, Stockholm, Sweden. [6] Division of Obstetrics and Gynecology, Karolinska Universitetssjukhuset, Stockholm, Sweden. ✉email: xfcheung@hku.hk

Congenital heart disease (CHD) is characterised by structural defects of the heart with or without associated abnormalities of the great arteries[1]. It is the most common birth defect that affects 3–4 in 10,000 live births[2]. Tetralogy of Fallot (TOF) is the most common cyanotic CHD and consists of four components, which include a ventricular septal defect (VSD), right ventricular (RV) outflow obstruction, RV hypertrophy, and an overriding aorta[3]. Despite surgical repair with closure of the VSD and relief of RV outflow obstruction, right[4] and left[5] ventricular dysfunction has been well documented in patients long-term after repair of TOF. Chronic RV volume overload due to chronic pulmonary regurgitation, dyskinesia of the aneurysmally dilated RV outflow, and adverse ventricular-ventricular interaction have been regarded as the major culprits[4,6–9]. Additionally, cardiac arrhythmia constitutes an important cause of late morbidity and mortality[10,11]. Risk factors for its development include an older age[10], significant pulmonary and tricuspid regurgitation[11], unfavourable electrophysiologic markers including prolongation of QRS duration and severe QRS fragmentation[12,13], a greater number of cardiac surgeries[14,15], and left heart disease[10]. It is therefore a common belief that ventricular dysfunction and cardiac arrhythmias in repaired TOF occur secondary to chronic alteration of cardiac load, adverse electrophysiologic substrates, and sequelae of open heart surgery. Whether primary intrinsic developmental, functional, and electrophysiological abnormalities of cardiomyocytes exist in TOF has not been explored.

About 80% of TOF cases are nonsyndromic with no generally identifiable genetic causes[16–18]. In isolated cases of nonsyndromic TOF, mutations of *NKX2-5*, *TBX1*, *HAND2* and *PITX2* have been reported[19–22]. Whole exome sequencing has further revealed that the *NOTCH1* locus is the most frequent site of genetic variants, followed by *FLT4*, predisposing to nonsyndromic TOF[18]. About 20% of TOF cases are associated with syndromal or chromosomal anomalies[23], with the majority (12–18%) having DiGeorge (DG) syndrome with 22q11.2 deletion[24–26]. In DG syndrome, *TBX1* is considered to be the major causal gene for the development of congenital cardiac anomalies[27,28]. While *Lgdel*/+ and *Tbx1*-/- mouse models have been shown to display cardiac and extracardiac phenotypes of DG syndrome, perinatal lethality of these mouse models precludes evaluation of cardiac functional and electrophysiologic perturbations[27,29].

Advances in human induced pluripotent stem cell (hiPSC) technologies have enabled the generation of patient-specific cardiomyocytes (hiPSC-CMs) for modelling CHD[30]. This is of particular relevance to the study of intrinsic developmental, functional, and electrophysiological abnormalities of cardiomyocytes in the setting of CHD, free of the confounding influence of structural abnormalities, abnormal haemodynamics, and surgical trauma. Based on single-cell transcriptomics of engineered cardiac tissues from patient-specific hiPSC-CMs, our group has recently demonstrated abnormal developmental trajectory and intrinsic contractile defects in hypoplastic right heart syndrome[31].

To explore possible intrinsic cardiac defects in TOF, we compared hiPSC-CMs and hiPSC-derived cardiac progenitors (hiPSC-CPs) from two TOF with DG (TOF-DG) syndrome patients, two TOF patients with no DG (TOF-ND) syndrome, and two healthy controls. We fabricated hiPSC-CMs into the human cardiac anisotropic sheet (hCAS), an anisotropic bioengineered tissue construct, for electrophysiological assessment, particularly of the risk of re-entrant arrhythmia. Single-cell RNA sequencing (scRNA-seq) was performed on hiPSC-CMs and in vitro cardiac progenitors (hiPSC-CPs). We demonstrated impaired ventricular specification in TOF-DG-hiPSC-CMs and identified a subgroup of TOF-DG-hiPSC-CMs that showed impaired cardiac gene expression and upregulated expression of

neural genes. We then showed the bifurcated cardiac differentiation, as marked by differential *RGS13* expression, of TOF-DG-hiPSC-CPs. Further electrophysiological assessment revealed increased arrhythmogenicity in TOF-DG-hCAS. These findings were found only in the TOF-DG but not TOF-ND patient-derived hiPSC-CMs and CPs, which have implications for the worse clinical outcomes of TOF-DG patients.

## Results

### Generation and characterisation of patient-specific hiPSCs and hiPSC-CMs.
hiPSC lines were established from two TOF-DG patients, two TOF-ND patients, and two healthy controls with pluripotency markers and germ layer markers verified (Supplementary Figs. 1 and 2). Whole genome sequencing confirmed, respectively, the presence and the absence of 22q11.2 deletion in the hiPSC lines from TOF-DG and TOF-ND patients (Supplementary Fig. 3). No missense or nonsense mutations of previously reported CHD-related genes were found from any of the hiPSC lines. Cardiac differentiation was accomplished through the APLNR+ sorting protocol developed by our laboratory[32]. All hiPSC lines showed satisfactory differentiation efficiency with over 70% TNNT2+ cells. On day 12 post-differentiation, hiPSC-CMs were fabricated into hCAS and allowed to mature for another 10 days before transcriptomic and functional assessments.

### TOF-DG patient-specific hiPSC-CMs exhibit defective ventricular specification.
We first determined the transcriptomic profile of differentiated hiPSC-CMs from patients and controls by scRNA-seq using the 10X Genomics platform. Clustering of 21,965 cells using Uniform Manifold Approximation and Projection (UMAP) identified 7 distinct Seurat clusters (Fig. 1A). Seurat clusters A0, A1, A2, A5 expressed relatively higher levels of cardiac genes including *NKX2-5*, myosin light/heavy chain, and troponin (Fig. 1B), which identified these clusters as cardiomyocytes. Cluster A5 expressed also higher levels of mitotic genes, suggesting the proliferative nature of the cardiomyocytes. Clusters A3 and A4 expressed relatively lower levels of cardiac genes (*NKX2-5*, *TNNT2*) but enriched in the expression of genes related to extracellular matrix (ECM) remodelling (Fig. 1B), the profile of which is compatible with cardiac fibroblasts. Clusters A6 and A7 were defined by endodermal lineage (*AFP*, *APOA1*, *APOA2*) and endothelial (*FLT1*, *ESAM*, *PLVAP*) markers, respectively (Fig. 1B). Hence, among the sequenced cells, the majority were cardiomyocytes (86%), some were cardiac fibroblasts (10%), and very few were endodermal (<1%) and endothelial derivatives (<1%).

With a focus on examining the cardiac transcriptome, we isolated hiPSC-CMs (clusters A0, A1, A2, A5) and performed clustering using UMAP (Fig. 1C). New Seurat clusters (B0, B1, B2, B3) were identified (Fig. 1C). hiPSC-CMs derived from patients and controls were unevenly distributed among clusters (Fig. 1D). Cluster B0 consisted of hiPSC-CMs almost exclusively (98%) derived from TOF-DG patients, cluster B1 consisted of cells mainly from a control subject and the remaining cells from TOF-DG patients, while clusters B2 and B3 consisted of hiPSC-CMs derived from TOF-ND patients and controls.

Cluster B0 was found to have upregulated expression of non-myocyte genes, with 4 out of 10 top upregulated genes being rarely (<1%) found in other clusters (Fig. 1E). These non-myocyte genes were non-endodermal or endothelial-related. The absence of enrichment in genes related to extracellular matrix remodelling also distinguished cluster B0 cells from cardiac fibroblast. Cluster B1 was enriched in the *IRX* gene family (*IRX1, IRX2, IRX3*), while cluster B2 was marked by *LBH* expression (Fig. 1E). The orthologs of *IRX* genes and *LBH* have been

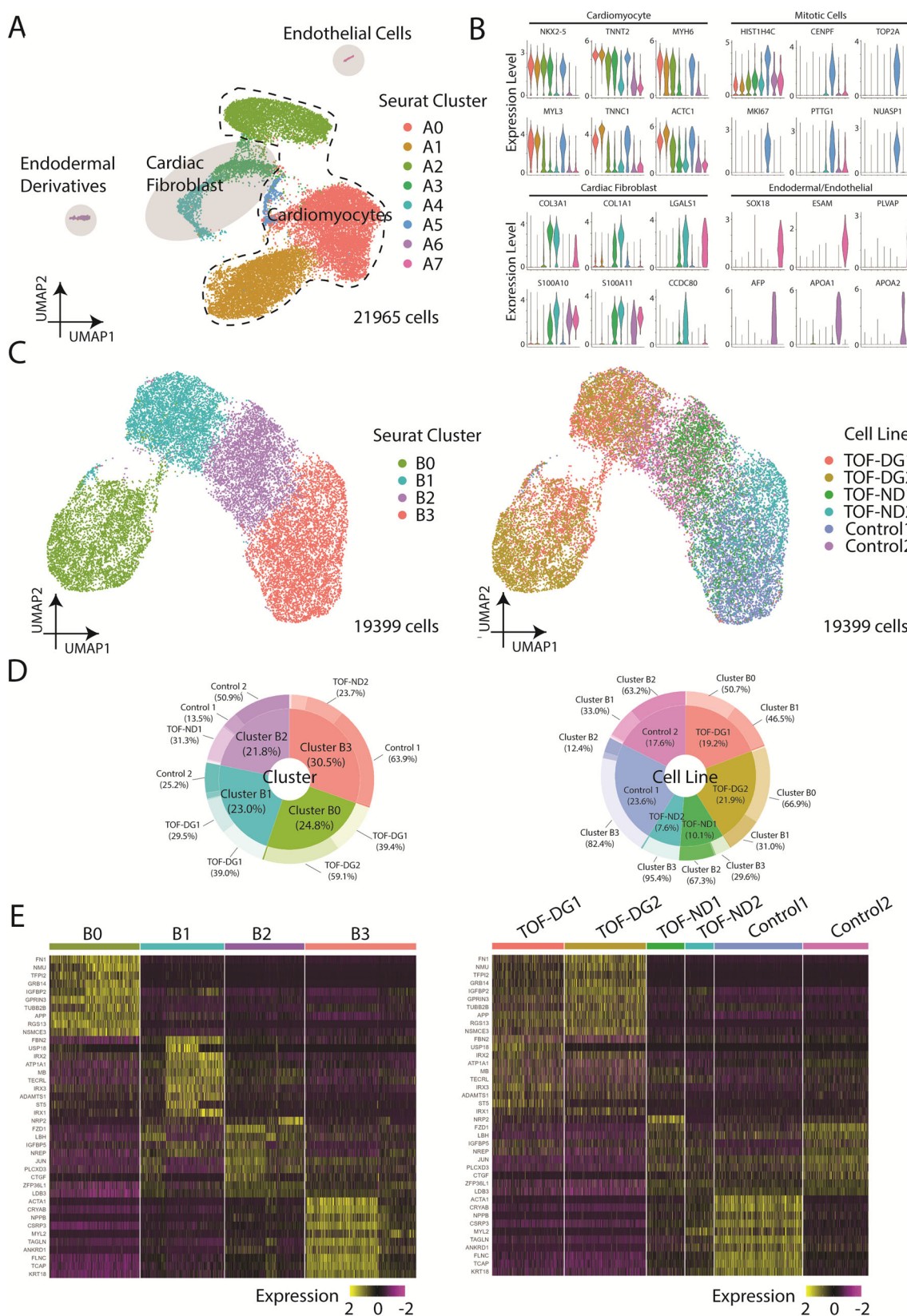

reported to be expressed in early in vivo differentiated murine cardiomyocytes[33,34]. Cluster B2, when compared with cluster B1, had upregulated expression of ventricular cardiomyocyte markers (*MYH7, NPPB*). Cluster B3 showed the highest expression of ventricular cardiomyocyte marker (*MYL2*) (Fig. 1E). The existence of clusters B1, B2 and B3 suggested progressive

specification of hiPSC-CMs towards a ventricular transcriptomic profile.

To further investigate the transcriptomic changes during ventricular specification, we performed pseudotime analysis on clusters B1, B2 and B3. Based on the enrichment in the *IRX* genes family, cluster B1 was manually selected as the root node for the

**Fig. 1 Overview of the single-cell transcriptome of hCAS (Day 22 post-differentiation). A** UMAP presentation of all the cell types identified in hCAS (TOF-DG, TOF-ND and control). Colouring: Seurat clusters; dashed line: cardiomyocytes; light grey: non-myocytes. **B** Violin plots of gene expressions of cardiomyocytes, mitotic cells and non-myocytes: cardiac fibroblast, endodermal derivatives and endothelial cells. Colouring: Seurat cell clusters as in (**A**). **C** UMAP presentations of all the hiPSC-CMs identified in hCAS (TOF-DG, TOF-ND and control). Data were coloured by Seurat clusters (left panel) and cell line (right panel), respectively. **D** Donut plots of the compositions in each Seurat cluster (left panel) and cell line (right panel), respectively. Data were calculated from hiPSC-CMs only. **E** Heatmap presentations of the top 10 upregulated genes expressed in each hiPSC-CMs-Seurat cluster. Data were grouped by Seurat cluster (left panel) and cell line (right panel), respectively.

pseudotime analysis (Fig. 2A). Along the trajectory, TOF-DG-hiPSC-CMs were mainly distributed in the beginning, whereas TOF-ND-hiPSC-CMs were mainly distributed in the middle to the end. Control-hiPSC-CMs could be found throughout the trajectory (Fig. 2A).

Genes with similar expression patterns were grouped into modules (Fig. 2B) and subjected to Gene Ontology (GO) enrichment analysis. Significant GO terms were found in both Module 1 and 2 (Fig. 2C). Module 1, showing high expression scores along the trajectory, was enriched in genes related to ventricular morphogenesis (*TNNI3, MYL2, MYH7*) and muscle contraction (*ACTC1, CSRP3, TCAP*). Module 2, showing decreasing expression scores along the trajectory, was enriched in genes related to extracellular matrix remodelling (*COL2A1, COL5A2, COL14A1*). The *IRX* genes family and *LBH* were also assigned to module 2, showing decreasing expression scores along the trajectory.

In sum, pseudotime analysis on hiPSC-CMs derived from the hCAS platform identified progressive ventricular specification with increased cardiac gene expression. Control- and TOF-ND-hiPSC-CMs were comparable in such progressive specifications. However, most of the TOF-DG-hiPSC-CMs retained the more primitive expression profile.

**A subset of TOF-DG patient-specific hiPSC-CMs showed downregulation of cardiac gene expression but upregulation of neural gene expression**. To further analyse the non-myocyte gene expression in cluster B0, differentially expressed genes (DEGs) were identified by comparing cluster B0 to each of the other clusters. More than 200 DEGs, upregulated or down-regulated, were identified from cluster B0 (Supplementary Data 1). GO enrichment analysis consistently identified significant GO terms related to cardiac muscle contraction among downregulated genes and those related to neural development among upregulated genes in cluster B0 (Fig. 2D). Therefore, cluster B0 differed from the rest of the clusters by down-regulated cardiac gene expression as well as upregulated neural gene expression. Yet, none of the above DEGs, including downregulated cardiac genes and upregulated neural genes, could be mapped to the haploinsufficient region in both TOF-DG hiPSC lines.

We further attempted to identify the corresponding transcription factors (TFs) and gene-regulatory networks from each of the clusters through Python implementation of the Single-Cell rEgulatory Network Inference and Clustering (pySCENIC). Among the top 10 TFs with higher activity in cluster B0, 4 were involved in neural development (*TCF3, SOX11, SOX4* and *ZEB1*) and 2 in cardiac and neural development (*MEF2C* and *MYEF2*) (Fig. 2E). The expression levels of five neural TFs (*SOX11, SOX4, ZEB1, MEF2C* and *MYEF2*) were upregulated in cluster B0 compared with those in other clusters (Fig. 2F). The finding of upregulated expression of these neural TFs was consistent with increased expression of neural genes in cluster B0. Similar to the DEGs, the above neural TFs were not mapped to the haploinsufficient region in both TOF-DG cell lines.

Therefore, a significant portion of hiPSC-CMs from TOF-DG (cluster B0) showed downregulation of cardiac genes and upregulation of ectopic neural genes. Failure to map the DEGs and TFs in cluster B0 to the 22q11.2 region suggested that this transcriptomic signature is probably an indirect consequence of the haploinsufficiency. Unlike the hiPSC-CMs from the control and the TOF-ND groups, which showed a continuum in the ventricular specification, cluster B0 formed a distinct cluster from the rest of the hiPSC-CMs from TOF-DG (cluster B1). The origin of cluster B0 was further explored as described below.

**Bifurcated cardiac differentiation of TOF-DG cardiac progenitors**. To understand the emergence of TOF-DG-cluster B0, we examined the transcriptomic profile of in vitro cardiac progenitors. Based on our previous study[32], second heart field (SHF) gene expression peaked on post-differentiation Day (D) 7 and 8. We therefore performed scRNA-seq on hiPSC-cardiac progenitors (hiPSC-CPs) on D7 and D8 and immature hiPSC-CMs on D11 from 2 controls (Control 1 and 2) and 2 TOF- DG (TOF-DG1 and 2) and 1 TOF-ND (TOF-ND1) patients (Fig. 3A, B). In total, 87834 cells were retained after excluding less than 1% of endodermal derivatives and endothelial cells. Pseudotime and GO enrichment analysis identified upregulation in cardiac gene expression and downregulation in transcription and translation and cell cycle activity along the trajectory (Supplementary Fig. 4A–D). Similar to our previous study[32], SHF gene expressions, including *ISL1, MEF2C, HAND2* and *FGF10*, were found in the sequenced time points (Fig. 3C). TBX1, one of the genes within the 22q11.2 microdeletion region, is also a marker of SHF. Among the sequenced time points, *TBX1* expression was found on Day7, with a variable proportion of *TBX1*[+] hiPSC-CPs (10–30%) found from different lines (Fig. 3D). However, the *TBX1* expression level was comparable between the D7-*TBX1*[+] hiPSC-CPs from different lines (Fig. 3D).

Time-matched comparison with controls only identified 20, 10 and 22 DEGs from D7-, D8- and D11-TOF-DG-hiPSC-CP/CMs with no significant GO terms found. None of these DEGs were mapped to the 22q11.2 microdeletion regions in the two TOF-DG-hiPSC lines. Ectopic gene expressions from TOF-DG-cluster B0 were then examined in the hiPSC-CPs and immature hiPSC-CMs. Unlike TOF-DG-cluster B0 (differentiated hiPSC-CMs), neural-related genes were not found in TOF-DG-hiPSC- CPs (D7 and D8) and D11-immature hiPSC-CMs. On the other hand, a subset of cells marked by *RGS13* expression, 1 of the top 10 upregulated genes in TOF-DG-cluster B0 (Fig. 1E), was consistently found in TOF-DG derived cells in all the sequenced time points (D7, D8 and D11) (Fig. 3E), but not in TOF-ND1 and the control groups.

Ectopic gene expression has been recently reported in the progenitors from *Tbx1* conditional null (*Tbx1*-cKO) mouse[35]. To compare our hiPSC-CP and immature hiPSC-CM data with the *Tbx1*-cKO mouse, we made a cross-species comparison with R package SingleCellNet[36]. Fourteen clusters were identified from the *Tbx1*-cKO mouse dataset (Supplementary Fig. 5A) with gene expression reported in the publication[35] (Supplementary Fig. 5B), including four different cardiac progenitors (multilineage progenitor, MLP; anterior SHF, aSHF; posterior SHF, pSHF and

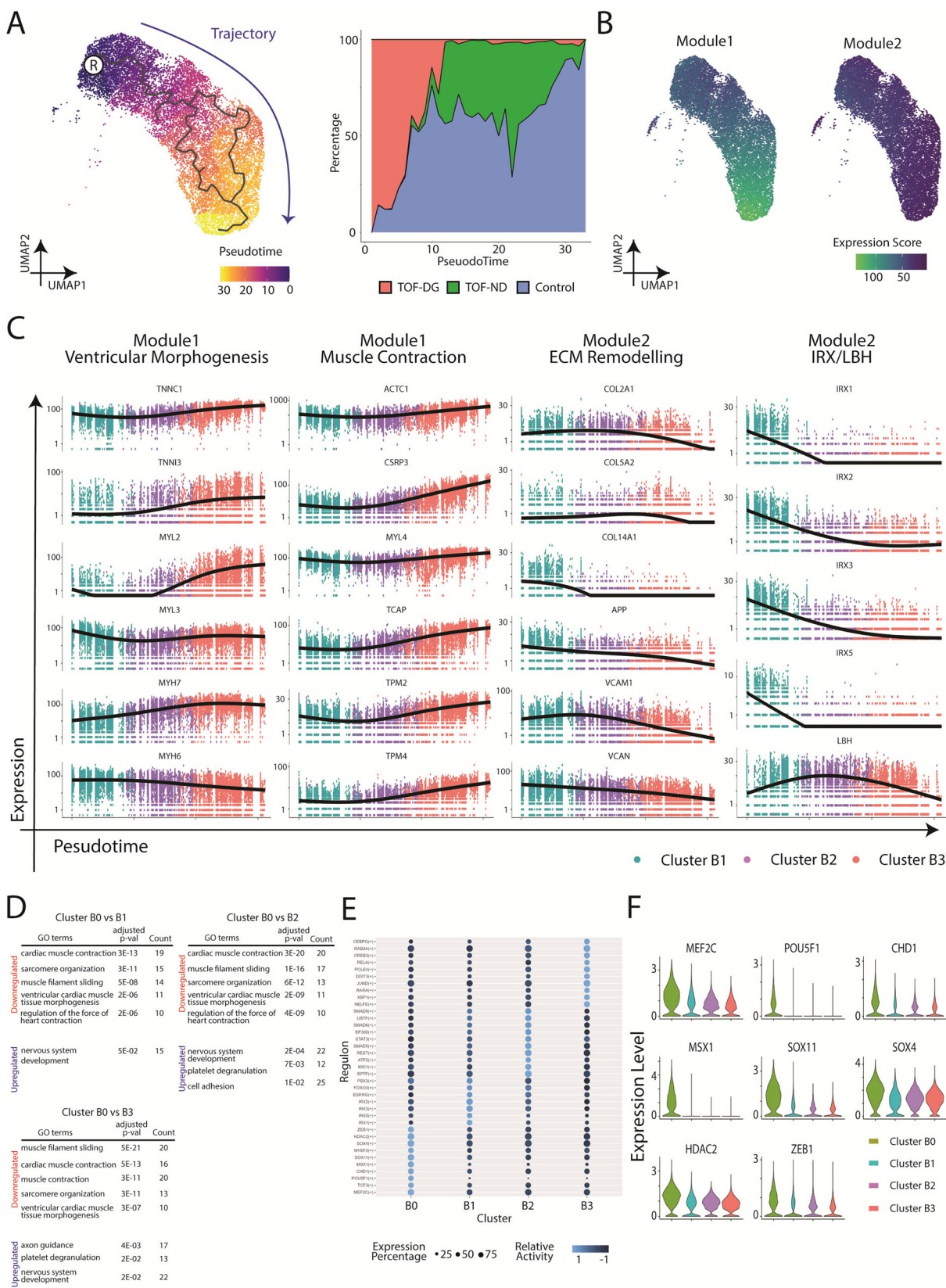

proepicardium; PEO) and cardiomyocytes. Our D7-hiPSC-CPs were mostly similar to aSHF whereas D11-hiPSC-immature cardiomyocytes were mostly similar to cardiomyocytes (Supplementary Fig. 5C). In the *Tbx1*-cKO mouse dataset, *Pax8*, the major ectopic gene reported by Nomaru et al.[35], was exclusively found in MLP and lung progenitors but not in aSHF and cardiomyocytes (Supplementary Fig. 5D). We did not identify *PAX8* expression in our D7-D11 dataset. Also, *RGS13*, the ectopic gene found in our D7-D11 dataset, was not identified in the *Tbx1*-cKO mouse. In other words, while ectopic gene expression was found in both *Tbx1*-cKO mouse and our DG-hiPSC-CPs/CMs, the identity of and the population expressing the ectopic gene are different.

**Fig. 2 Trajectory, GO enrichment and regulon analysis of hCAS-hiPSC-CMs. A** Pseudotime analysis of hCAS-hiPSC-CMs. UMAP coordination and gene expressions of cluster B1, B2 and B3 (Fig. 1C) were extrapolated for Monocle3 calculation. Root node was denoted with ®. Trajectory was coloured in blue. **B** Gene modules serve as the function of the pseudotime analysis and trajectory inference of hCAS-hiPSC-CMs. **C** Gene expression dynamics along the trajectory. Ventricular morphogenesis, muscle contraction and ECM remodelling were the GO terms enriched in the gene modules. X-axis: Pseudotime ; Y-axis: Expression Level. ECM extracellular matrix. **D** GO enrichment analysis of cluster B0 from hCAS-hiPSC-CMs. GO terms found to be enriched in the downregulated/upregulated genes from cluster B0 (compared against cluster B1/B2/B3) were shown. P-values were adjusted for Bonferroni multiple comparisons. **E** Regulon analysis of hCAS-hiPSC-CMs. The top regulons (at most 10) with higher relative activity in each Seurat cluster were shown in the dot-heatmap plot. **F** Gene expression of the top regulons (identified as DEGs) from cluster B0. Gene expressions were shown in the violin plot.

**Altered electrophysiological parameters and increased arrhythmogenicity in TOF-DG-hiPSC-CMs.** In addition to transcriptomic profiling, we explored the electrophysiology and arrhythmogenicity of the differentiated hiPSC-CMs using our hCAS platform to provide a more comprehensive assessment of the hiPSC-CMs. Electrophysiological parameters, including action potential (AP), calcium handling and effective refractory period (ERP), were measured under electrical pacing at 1 Hz.

Compared with both controls, TOF-DG2-hCAS showed significant shortening of AP duration (50 and 90% to repolarization, APD50 and APD90) (Fig. 4A). Similarly, when compared with both controls, TOF-DG2-hCAS showed significant shortening of AP upstroke time and time to decay (50 and 90% from peaks) (Fig. 4A). Hence, the AP characteristics differed significantly between TOF-DG2 and controls.

Calcium handling was assessed with calcium transient (CaT) upstroke time and time to decay (50 and 90% from the peak). Significant shortening of CaT upstroke time was found in TOF-DG2-hCAS, whereas significant shortening of time to decay (50% from the peak) was found in both TOF-DG2-hCAS and TOF-ND1-hCAS when compared with both controls (Fig. 4B).

We further assessed the ERP of the hiPSC-CMs, the shortening of which predisposes to the development of cardiac arrhythmias. Significant shortening of EPR was found in TOF-DG2-hCAS, while prolongation of ERP was found in TOF-ND2-hCAS (Fig. 4C). Importantly, TOF-DG2-hCAS showed a significantly higher incidence of re-entry arrhythmia during programmed electrical pacing (PES) (Fig. 4D). A representative AP tracing with the isochrone map of re-entry arrhythmia from TOF-DG2-hCAS is shown in Fig. 4E.

Whereas TOF-DG2-hCAS showed significant shortening of AP duration, CaT upstroke time and time to decay, and ERP, TOF-DG1-hCAS was similar to controls in terms of these electrophysiological parameters assessed (Fig. 4A–D). We therefore further explored the transcriptomic differences between TOF-DG1 and TOF-DG2 in cluster B0. Compared with cluster B0-TOF-DG1, cluster B0-TOF-DG2 showed downregulation of cardiac genes related to muscle contraction (Fig. 4F). While these cardiac genes were downregulated in both cluster B0-TOF-DG1 and cluster B0-TOF- DG2 when compared with other clusters (B1, B2 and B3) (Fig. 4F), their extent of downregulation was more prominent in cluster B0-TOF-DG2 (Fig. 4F). Furthermore, *PKP2*, its mutation being reported to be associated with arrhythmogenic RV dysplasia[37], was found to be among the downregulated cardiac genes in cluster B0-TOF-DG2 (Fig. 4F). The varying extent of downregulation of cardiac genes may hence account for the functional discrepancy observed among cell lines from different patients in the hCAS platform.

## Discussion

In this study, we discovered impairment of ventricular specification, downregulated expression of cardiac genes, and upregulated expression of ectopic neural genes in TOF-DG-hiPSC-CMs, bifurcated cardiac differentiation of TOF-DG cardiac progenitors,

and increased arrhythmogenicity in hCAS derived from TOF-DG-hiPSC-CMs in the absence of structural defects. By contrast, TOF-ND-hiPSC-CMs and -CPs were comparable to control-hiPSC-CMs and -CPs in terms of ventricular specification, differentiation of cardiac progenitors, and arrhythmogenicity.

Studies on TOF patient-specific hiPSC-CMs are limited. Only two reports have to date studied TOF-ND-hiPSC[38,39], and one studied TOF-DG-hiPSC[40] with varying objectives and approaches. Using bulk-RNA sequencing of differentiated TOF-ND-hiPSC-CMs, Kitani et al. reported 250 DEGs in TOF-ND-iPSC-CMs but with no identifiable enriched biological processes[38]. Using the only available TOF-ND-hiPSC cell line, after discarding cell lines with somatic mutation, Grunert et al. reported DEGs between a TOF-ND patient and healthy father cell lines based on bulk-RNA sequencing with enrichment in cardiac system-related GO terms[39]. On the other hand, to address the question as to how Tbx1 represses *Mef2c* expression, Pane et al. performed quantitative real-time polymerase chain reaction (qRT-PCR) on the TOF-DG-hiPSC-CPs and indeed found upregulation of *MEF2C* in TOF-DG-hiPSC-CPs[40]. Hence, the existing data are heterogeneous with regard to objectives and cardiac differentiation methodology and limited to the utilisation of low-resolution transcriptomic tools for assessing heterogeneous cell populations. By contrast, the strengths of the present study include a direct comparison of syndromic and nonsyndromic TOF with the hiPSC model, adoption of a consistent cardiac differentiation methodology, and utilisation of high-resolution scRNA-seq to determine transcriptomic differences between TOF-DG versus TOF-ND hiPSC-CPs and CMs. Furthermore, the ability to remove transcriptomic data from the non-myocyte population and the non-cardiac lineage has enabled focused analysis and interpretation of data pertaining only to hiPSC-CPs and CMs. Additionally, we assessed the hitherto unexplored aspect of electrophysiology of syndromic and nonsyndromic TOF-hiPSC-CMs and unveiled increased arrhythmogenicity of TOF-DG-hiPSC-CMs.

Based on scRNA-seq of hiPSC-CPs and their further categorisation according to *RGS13* expression, we discovered that TOF-DG-hiPSC-CPs bifurcated as early as D7 during in vitro cardiac differentiation. *RGS13*[+] TOF-DG-hiPSC-CMs further showed downregulation of cardiac gene expression and ectopic expression of neural genes after prolonged culture to Day22. Importantly, the bifurcated differentiation and ectopic expression of neural genes were unique to TOF-DG-hiPSC-CMs, which implicates their relationship with the deletion of 22q11.2. In DG syndrome, *TBX1* is regarded as the major gene that causes congenital cardiac anomalies. Recently, using the *Tbx1*-conditional null (*Tbx1-cKO*) mouse model, Nomaru et al. showed that *Tbx1* is expressed in the early mesodermal progenitors and governs their differentiation to cardiac progenitors[35]. *Tbx1* knockout in the early mesodermal progenitors was shown to cause downregulation of genes involved in cardiac muscle development and ectopic non-myocyte expression of genes involved in axonogenesis and inner ear development, mediated partly through alteration of chromatin accessibility[35]. Our hiPSC model of TOF-DG captured similar transcriptomic signature in the cardiac lineage, including

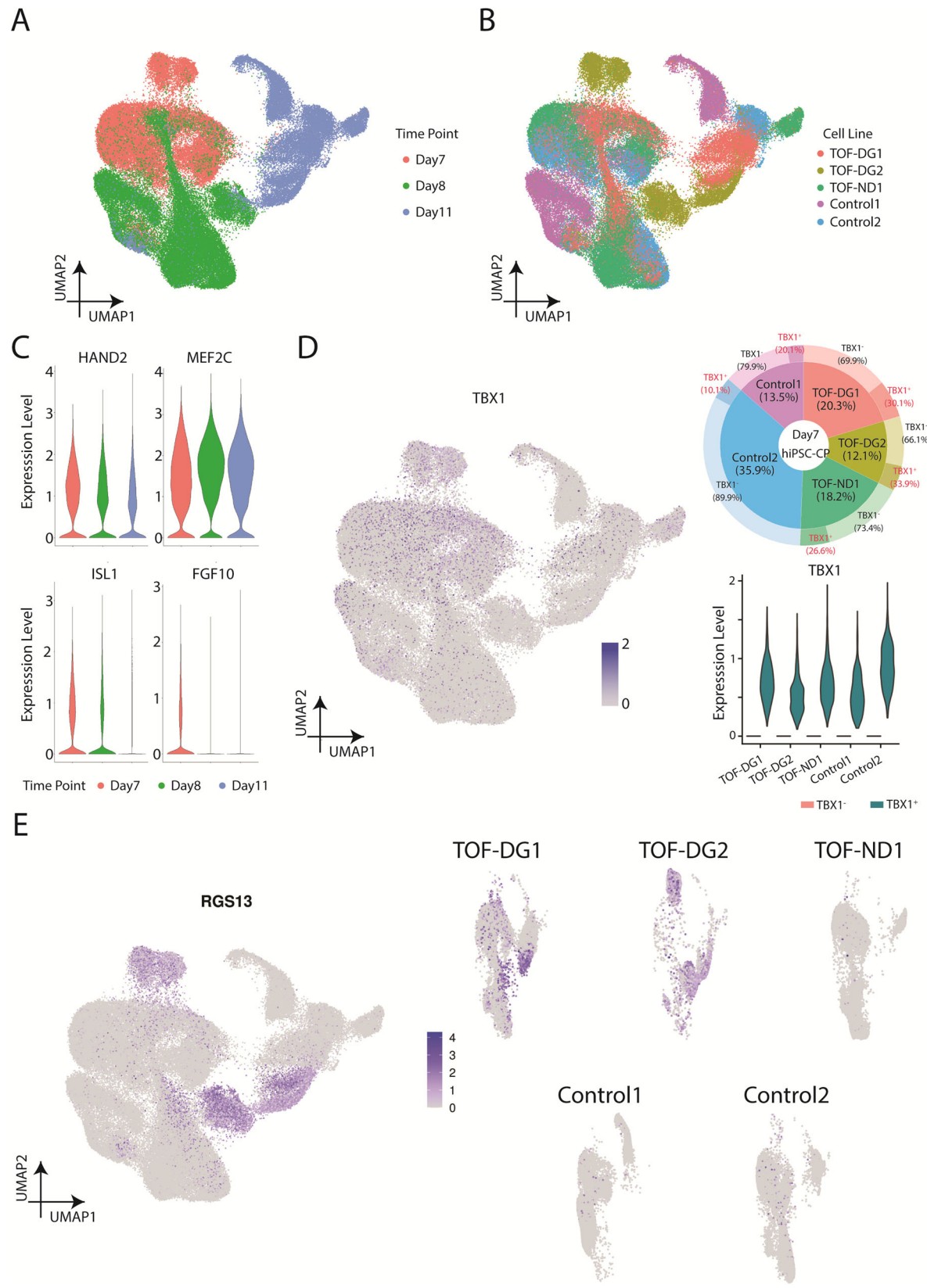

downregulation of cardiac gene expression and ectopic non-myocyte gene expression. However, our findings differ from the *Tbx1-cKO* mouse in the identity of the ectopic gene (*RGS13* vs *Pax8*) and the reported population (hiPSC-CPs and CMs vs MLP). This may suggest that the mechanism behind the

transcriptomic features reported in the current study is more complicated than just TBX1 haploinsufficiency.

In fact, the trans-regulatory effect of 22q11.2 microdeletion was found in DG primary cell lines[41]. Zhang et al. reported multi-layer epigenomic changes due to 22q11.2 microdeletion, ranging

**Fig. 3 Overview of the single-cell transcriptome of Day7 and Day8 hiPSC-CPs and Day11 immature hiPSC-CMs. A** UMAP presentations of all the hiPSC-CPs and hiPSC-CMs (TOF-DG, TOF-ND and control). Data were coloured by sequenced time point. **B** UMAP presentations of all the hiPSC-CPs and hiPSC-CMs (TOF-DG, TOF-ND and control). Data were coloured by cell line. **C** Violin plots of SHF gene expression, including *ISL1, MEF2C, HAND2* and *FGF10*. **D** *TBX1* expression in hiPSC-CPs. *TBX1* expression on Day7/8/11 in vitro cardiac differentiation was shown in the UMAP plot. Highest *TBX1* expression was found in D7-hiPSC-CPs, and the proportion of the *TBX1*+ was shown in donut plot and marked with a red label. *TBX1* expression in *TBX1*+ D7-hiPSC-CPs was shown in violin plot. **E** UMAP plots of *RGS13* expression on Day7/8/11 in vitro cardiac differentiation. *RGS13* expression were presented with all the cell lines together (left panel) and split into every individual cell line (right panel).

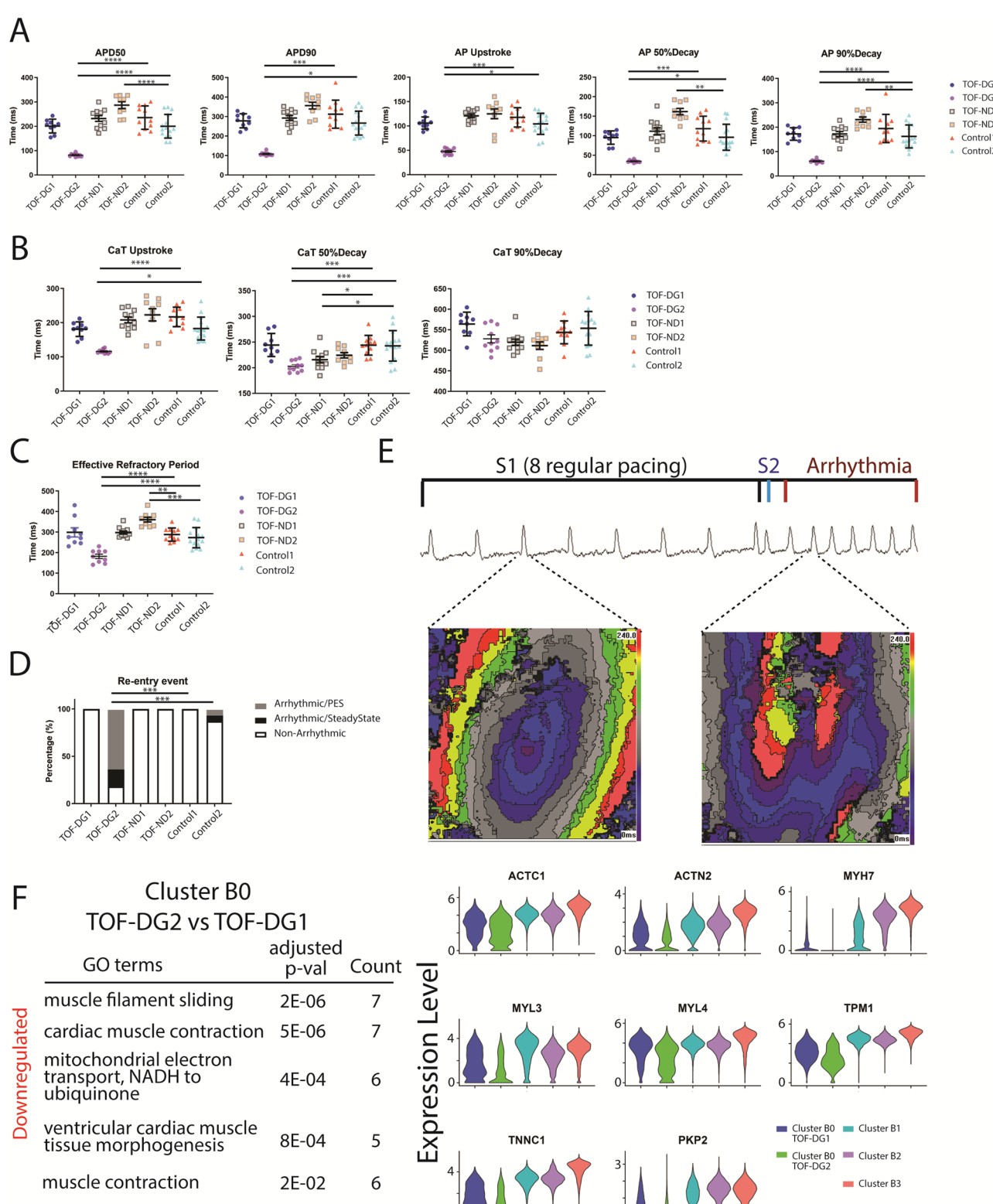

**Fig. 4 Electrophysiological assessment of hCAS. A** Dotplots of action potential (AP) characteristic. APD action potential duration. **B** Dotplots of calcium transient (CaT) characteristic. **C** Dotplot of the effective refractory period. Data were presented in mean ± SD for Fig. 1A–C. Statistical test: Ordinary one-way ANOVA followed by Tukey's multiple comparisons test (APD50, AP90%Decay, CaT 50%Decay, CaT Upstroke and effective refractory period); Kruskal–Wallis test followed by Dunn's multiple comparison test (APD90, AP upstroke, AP50%Decay and CaT 90%Decay). Significant findings between TOF(DG/ND) and control were marked by asterisks. * $P < 0.05$; ** $P < 0.01$; *** $P < 0.001$; **** $P < 0.0001$. **D** Percentage of arrhythmic re-entry event of TOF (DG/ND) and control hCAS during steady-state pacing (SteadyState) and programmed electrical stimulation (PES). Statistical test: Fisher's exact test. *** $P < 0.001$. **E** Representative action potential (AP) tracing and isochrone map of re-entry event during PES from TOF-DG2. The upper panel showed the representative AP tracing during PES (S1S2) followed by the incidence of arrhythmic re-entry event. The lower panel showed two isochrone maps which correspond to a normal AP (left) and an arrhythmic AP (right). **F** Cardiac gene downregulation in cluster B0-TOF-DG2. GO enrichment analysis was performed on the comparison between cluster B0-TOF-DG2 and cluster B0-TOF-DG1. GO terms enriched in the downregulated genes are shown in the table (left), and the corresponding genes are shown in the violin plot (right).

from histone modification to global chromosomal topological changes[41]. The epigenomic analysis suggests that copy number changes of the genes within a large copy number variant are not sufficient to explain all the genetic expression changes observed[41]. In our study, the majority of the DEGs identified, including the downregulated cardiac genes and the upregulated non-myocyte genes, are located outside of the 22q11.2 microdeletion region. Investigating the chromosomal changes along the in vitro cardiac differentiation would provide more insight, especially on the consistent upregulation of *RGS13* from hiPSC-CPs to hiPSC-CMs in TOF-DG.

From the clinical perspective, there are accumulating data to suggest that TOF-DG patients may have worse early and late cardiovascular outcomes compared with TOF-ND patients[42,43]. In a retrospective review of the early surgical outcomes of 208 TOF (44 with DG) patients, Mercer-Rosa et al. found that DG compared with ND patients required a longer duration of cardiopulmonary bypass time and intensive care and speculated that DG may affect genes contributing to RV function, adaptation to surgical stress, and postoperative RV restrictive physiology[42]. With regard to long-term outcomes, based on a large cohort of patients with TOF with and without pulmonary atresia registered in the Dutch nationwide CONgenital CORvitia (CONCOR) registry, Kauw et al. found significantly decreased long-term survival of adult TOF-DG patients compared to TOF-ND patients (76% versus 89%) at 12 years of follow up, although the mechanism was unclear[44]. Nonetheless, a cross-sectional study of paediatric and adolescent TOF patients provided clues of worse cardiac performance in DG patients compared with ND patients, as evidenced by the lower RV cardiac index, RV stroke volume, and left ventricular ejection fraction, the need for one or more cardiac medication, cardiac-related hospitalisations, and maximum oxygen consumption on exercise testing[43]. Furthermore, among DG patients, including those without major congenital heart disease, sudden cardiac death, which may be related to cardiac arrhythmias, has been documented to be the most common cause of mortality[45]. The findings of this report hence shed important light on potential mechanisms of worse clinical outcomes as described above. Impaired ventricular specification and downregulated cardiac gene expression, which may affect myocardial function at rest and during stress, were evident in TOF-DG-hiPSC-CMs in the absence of any structural defects or remodelling secondary to altered haemodynamics. The altered electrophysiological characteristics and their predisposition to arrhythmias in the TOF-DG-hiPSC-CM line with the most downregulated cardiac gene expression (TOF-DG2), may also provide an explanation for sudden cardiac death in DG patients, even in the absence of cardiac malformation.

Several limitations to this study require discussion. Our hiPSC-CMs cultured using the hCAS platform showed varying degrees of ventricular expression, which may affect electrophysiologic assessment. Nonetheless, this applied to all cell lines and is unlikely to explain the

transcriptomic and functional defects in the TOF-DG-hiPSC-CM cell lines. Although prolonged culture (40–200 days) has been suggested to facilitate ventricular expression and promote cardiac maturation[46,47], this is not feasible with the hCAS platform as the seeded hiPSC-CMs would fall off. Revamping the engineered tissue construct is required in the future to further promote the degree of cardiac maturation. Finally, in vivo cardiac development and morphogenesis of the structural anomalies of TOF cannot be replicated using our model. Temporal and spatial interaction between cardiac progenitors, non-myocyte progenitors, and endothelial and other cells is crucial in the pathogenesis of TOF. Recently, self-organising cardioids from hiPSCs that intrinsically specify and morph into chamber-like structures containing a cavity have been developed, which may provide a more powerful platform to dissect the pathogenesis of CHD[48]. Given the bifurcated cardiogenesis as reported in this study, future experiments using the cardioids to focus on the relationship between 22q11.2 microdeletion and *RGS13* may increase our understanding of the defective cardiogenesis in TOF-DG syndrome.

## Methods

**Patients and generation of hiPSC.** We recruited two patients with TOF in association with DG syndrome (TOF-DG), two patients with nonsyndromic TOF (TOF-ND) and two healthy subjects in this study. The clinical phenotypes of all the subjects are shown in Supplementary Table 1. To generate hiPSCs, CD34+ peripheral blood mononuclear cells were collected from the whole blood samples of the recruited subjects with SepMate™ (STEMCELL Technologies), Lymphoprep™ (STEMCELL Technologies) and StemSpan™ H3000 (STEMCELL Technologies) with CC100 (STEMCELL Technologies). The isolated CD34+ mononuclear cells were cultured for 3 days before episomal reprogramming. Episomal reprogramming was achieved with nucleofection of pCXLE-hUL, pCXLE-hSK and pCXLE-hOCT3/4-shp53 using the Human CD34 Cell Nucleofector™ Kit (Lonza). Putative hiPSCs were cultured and expanded for 2 weeks in StemFlex™ Medium (SF, Gibco) on Geltrex™ (Gibco) coated 6-well plates. After expansion, hiPSC purification was performed with human Anti-TRA-1–60 Microbeads (Miltenyi Biotec). The purified hiPSCs were validated by immunofluorescence staining of pluripotency markers (OCT3/4, SOX2, SSEA4 and TRA-1–81) in hiPSCs (Supplementary Fig. 1) and germ layer markers (AFP for endoderm, a-SMA for mesoderm and TUB-b-III for ectoderm) in spontaneous embryoid body-formation assay (Supplementary Fig. 2).

The validated hiPSCs were maintained on Geltrex™ (Gibco) coated 6-well plates in SF medium, with the medium being changed every 2 days. Passage of hiPSCs was performed when confluency reached 80% (3–4 days in culture). After aspiration of the medium, hiPSCs were washed with 1 ml Dulbecco's phosphate-buffered saline (DPBS) (Gibco) followed by 3–5 min incubation with 500 µl of STEMPRO ACCUTASE (Accutase, Gibco) at 37 °C.

The dissociation was stopped by adding DMEM-F12 (Gibco) in a 2.5:1 ratio to Accutase. The dissociated cells were then resuspended in SF medium with 10 μM Y27632 (BioGems) and redistributed to a 6-well plate in a 1:6–1:10 ratio. The Y27632 containing SF medium was then replaced with fresh SF medium in 24 h.

**In vitro cardiac differentiation.** hiPSCs were ready for differentiation when the confluency reached near 80% on the 6-well plate. On day 0, single cells were collected from the enzymatic dissociation same as cell passaging. Single cells were then resuspended in SF medium with 10 μM Y27632, 1 ng/ml BMP4 (Gibco) and 40 μg/ml Matrigel™(Corning) and cultured in the ultra-low attachment 6-well plates (Corning). Cardiac differentiation took place in a hypoxic incubator (37 °C, 5% $O_2$) from day 0 to day 5 (before cell sorting) and in a normoxic incubator from day 5 (after cell sorting).

On day 1, small embryoid bodies (Ebs) were formed and collected into a 15 ml centrifuge tube (Corning) and centrifuged at $200 \times g$ for 1 min. The supernatant was removed, and the Ebs collected were resuspended in 3 ml DPBS followed by centrifugation at $200 \times g$ for 1 min. The supernatant was removed and the Ebs were collected and resuspended in StemPro™-34 SFM (SP34; Gibco) with 2 mM GlutaMAX™ Supplement (Gibco), 10 ng/ml ActivinA (Gibco), 10 ng/ml BMP4 and 50 μg/ml L-ascorbic acid (AA; Sigma-Aldrich). The Ebs were then transferred back to the original ultra-low attachment 6-well plate. On day 4, Ebs were collected into a 15 ml centrifuge tube and centrifuged at $100 \times g$ for 1 min. After removing the old medium and resuspension in 3 ml DPBS, Ebs were centrifuged at $100 \times g$ for 1 min. After removing DPBS, Ebs were resuspended in SP34 medium with 2 mM GlutaMax™, 5 μM IWR-1 (STEMCELL Technologies) and 50 μg/ml ascorbic acid. The Ebs were then transferred back to the original culture plate. On day 5, Ebs were collected in a 15 ml centrifuge tube and centrifuged at $100 \times g$ for 1 min. The old medium/supernatant was collected as the conditioned medium. Ebs were washed with 3 ml DPBS and centrifuged at $100 \times g$ for 1 min. After removing DPBS, Ebs were dissociated in 4 ml TrypLE™ Express Enzyme (1X) (TrypLE, Gibco) for 15 min in a shaking water bath at 37 °C. Single cells were collected by filtering the debris with a 40 μm cell strainer (Corning) after the dissociation. The dissociation was then stopped by adding 8 ml DMEM/F12 (Gibco). The supernatant was removed after centrifugation at $300 \times g$ for 3 min. After washing with DPBS, the cell number was determined by a hemocytometer (Abcam). Cells in DPBS were centrifuged at $300 \times g$ for 3 min again, and remove the DPBS. Buffer solution consisting of Hank's Balanced Salt Solution (1X) (HBSS; Gibco) with 0.5% Bovine Serum Albumin (BSA, Sigma-Aldrich) (HBSS/BSA) replaced DPBS for the subsequent steps. The single cells washed were then incubated in HBSS/BSA with human APJ Antibody (R&D; MAB8561; 1:200) for 30 min at room temperature. The total volume of the immunostaining used was based on the cell number (1 ml for 20 million cells). The single cells were resuspended every 10 min to prevent aggregation at the bottom. After incubation, cells were subjected to 2 rounds of centrifugation at $300 \times g$ for 3 min and washing with HBSS/BSA. After removing the HBSS/BSA, cells were incubated with goat anti-mouse IgG microbeads (Miltenyi Biotec, 1:5) in 100 μl HBSS/BSA for 15 min at room temperature. Again, the single cells were resuspended at 7 min to prevent cell aggregation. APLNR⁺ cardiac progenitors were then sorted out by MS columns (Miltenyi Biotec) and OctoMACS™ Separator (Miltenyi Biotec). APLNR⁺ cardiac progenitors were then cultured on Matrigel™ coated 6-well plates (1.5–2 × 10⁶ cells/well) in a conditioned medium 2 ml/well for 2 days. Thereafter, the sorted progenitors

switched from EB culture to monolayer culture at this moment. On Day 7, the old medium was replaced with 2 ml RPMI1640 with GlutaMAX™ (Gibco) with 2% B27 supplement (Gibco) (RPMI/B27). The medium was changed with fresh RPMI/B27 every other day (Supplementary Fig. 6). On Day 10/11, cells were dissociated for quality check assessed by fluorescence-activated cell sorting (FACS) flow cytometry.

**Flow cytometry.** hiPSC-CMs were dissociated into single cells by 0.025% Trypsin-EDTA (Gibco) (TE). Dissociated single cells were subjected to fixation and permeabilization with BD CytoFix/Cytoperm™ (BD Biosciences). The fixed and permeabilized samples were then incubated with 1% goat serum in PBS⁻/⁻ at 4 °C for 1 h. The samples were then stained with anti-Cardiac Troponin T antibody (Abcam, ab8295,1:400) in 1% goat serum/PBS⁻/⁻ overnight at 4 °C. After washing with PBS⁻/⁻ twice, samples were then stained with FITC conjugated rat anti-mouse IgG1 antibody (BioLegend; 406605,1:50) at 4 °C for 1 h. Quantitative assessment was performed with FACSCanto™ II (BD Biosciences). An example of flow cytometry analysis on the cTNT expression was included in Supplementary Data 2.

**Cardiac anisotropic sheets (CAS) fabrication and measurement.** The protocol was based on that described by our group previously[31] and modified to allow better cell survival during measurement. A 15-mm circular nanopatterned substrate (Shrinky Dinky) with microgrooves 10 × 5 × 5 μm (ridge×depth×width) was cast at 180 °C and treated with ultraviolet light and ozone (UVO-Cleaner, JeLight Company, Irvine, CA) for 15 min. The nanopatterned substrate was coated with Matrigel™ overnight in a 4-well plate (Thermo Fisher Scientific) before use. Day12 hiPSC-CMs (with cTNT% >70) were dissociated into single cells by 60 min incubation with collagenase type IV (Gibco) (200 U/ml, in DMEM/F12) and subsequent 10 min incubation with 0.025%TE. Cell dissociation was stopped by 4 volumes of 20% fetal bovine serum (FBS, Gibco) in DMEM/F12. Cell debris was removed with a 40 μm cell strainer before seeding. Dissociated single cells were resuspended in RPMI/B27 and seeded on the Matrigel™-coated nanopatterned substrate (1.3 million per substrate). The culture medium (1 ml RPMI/B27 per CAS) was changed daily until measurement.

For electrophysiological assessment using optical mapping, CAS was incubated with the potential-sensitive probe 10 μM RH237 (Invitrogen) and calcium indicator 2.5 μM Rhod-2, AM (Invitrogen) in DMEM/F12 (500 μl/CAS) for 30 min at 37 °C. Pluronic F-127 (Sigma-Aldrich) was used to dissolve Rhod-2, AM. After washing with DMEM/F12, CAS was then incubated with 50 μmol/L blebbistatin (Sigma-Aldrich) in DMEM/F12 at 37 °C for 10 min. After washing with DMEM/F12, CAS was maintained in 250 μl DMEM/F12 at 37 °C by a thermal plate (Tokai Hit, Shizuoka, Japan). Fluorescence lighting was then applied with a halogen light to CAS. External pulse stimulation was controlled by Master9 (AMPI, Jerusalem, Israel) via a unipolar point stimulation electrode (Harvard Apparatus, Holliston, MA). During steady-state pacing, pulses with 10 V, 10 ms duration and specific frequency (0.5, 1, 1.5, 2, 2.5, 3 Hz) were delivered to CAS. During programmed electrical training, 8 pulses of 1.5 Hz were first delivered, followed by an extra pulse at a predetermined interval. Live imaging at a sampling rate of 200 Hz was recorded with MiCAM Ultima (SciMedia, Costa Mesa, CA). The recorded images were analysed with BV Ana imaging software (SciMedia). Numerical data are included in Supplementary Data 3.

**Whole genome sequencing (WGS).** Genomic DNA was extracted from the frozen vials of hiPSC lines directly by DNeasy®

Blood & Tissue (QIAGEN) according to the manufacturer's protocol. Sequencing libraries were constructed by KAPA Hyper Prep Kit (KR0961-V1.14). WGS was performed using Illumina NovaSeq 6000 with paired-end 151 bp reads at 30X raw coverage.

Sequence quality was assessed by MultiQC[49]. Adaptor sequences were removed with Trimmomatic (version 0.39)[50], and trimmed reads were mapped to reference genome hg38 using bwa mem[51]. Aligned reads were then sorted and indexed using SAMtools[52].

For single nucleotide variant (SNV) calling, we applied GATK (version 4.2.0.0)[53] by following the GATK Best Practices[54], including duplicate reads removal and base quality score recalibration. Variants were called using the HaplotypeCaller mode of GATK, followed by joint genotyping across all six samples simultaneously. Variants were then filtered using variant quality score recalibration and hard filtering (root mean square mapping quality MQ < 50) in GATK. Further filtering was performed with VCFtools (version 0.1.16)[55] based on variant and genotype quality scores (Q < 20, GQ < 20), depth (DP < 4, DP > 50), and minor allele frequency (maf < 0.1, maf > 0.9). The total number of biallelic SNVs, allowing a maximum of three missing genotypes, was 4.2M. Annotation of SNV was performed using ANNOVAR 2020-06-07[56] with ClinVar (20210501) database. Only pathogenic and likely pathogenic SNVs were retained.

For structural variations, we performed screening on pooled control samples and TOF(DG/ND) samples using lumpy (version 0.2.13)[57] and GRIDSS (version 2.13.0)[58] with default parameter settings, followed by manual validation based on sequence coverage in Integrative Genomics Viewer (IGV)[59]. Runs of homozygosity regions (RoHs) were detected using Bcftools/Roh[60] using the GATK called SNVs. Mean Roh value was calculated for each group using sliding window length of 1Mbp and slide size of 25kbp.

**Single-cell RNA sequencing (scRNA-seq)**. Day 7 and 8 hiPSC-cardiac progenitors and day 11 hiPSC-CMs and hCAS were dissociated with 0.025TE for 10 min at 37 °C. Single cells were collected after debris removal with a 40 μm cell strainer. Live cells were resuspended in RPMI/B27 (1 million cells/ml).

Single-cell encapsulation and cDNA libraries were prepared by Chromium Next GEM Single Cell 3′ Reagent Kit v3.1 and Chromium Next GEM Chip G Single Cell Kit. Sequencing was performed using Illumina NovaSeq 6000 with paired-end 151 bp reads. Raw read alignment and filtering were performed with Cell Ranger (10X Genomics). Downstream analysis of scRNA-seq was performed with R-package Seurat3.0[61] except for the trajectory and regulon analysis, which was performed with the R-package Monocle3[62,63] and the python package pySCENIC[64]. Standard workflow from Seurat was adopted. Raw output from Cell Ranger (files in .tsv format. containing the expression matrix, the barcode and the gene names) was loaded into the R environment with Seurat. Cells with mitochondrial contents above 20–25% of the mapped reading or the low number of RNA transcripts (<500) were removed. After logarithmic normalisation, data were regressed with the expression score of the cell cycle gene expression. The batch effect in hCAS was regressed with R-package Harmony[65]. In addition, principal component analysis (PCA) and Uniform Manifold Approximation and Projection (UMAP) reduction were performed. Gene expression in violin plot and UMAP representations were generated with the function VlnPlot() and FeaturePlot(), respectively.

Differentially expressed gene (DEG) analysis was performed by the function FindMarkers() with MAST as the statistical test. Genes with expression in at least 25% of the cells in either group

of comparison, logarithmic fold change >0.5 and an adjusted P-value < 0.05 were regarded as DEGs. Gene-Ontology (GO) enrichment analysis was performed with DAVID Functional Annotation Bioinformatics Microarray Analysis[66]. Overrepresented GO terms were identified with Bonferroni-adjusted P < 0.05. DEGs and the corresponding GO term analysis can be found in Supplementary Data 1. The PCA and UMAP reduction calculated from Seurat was subsequently used in Monocle3 for trajectory analysis. The downstream trajectory analysis followed the standard workflow of Monocle3.

For cross-species comparison between *Tbx1*-cKO mouse dataset and our hiPSC-CPs/immature CMs, *Tbx1*-cKO mouse dataset was downloaded from GEO repository (GSE167493), including Mesp1 Cre E9.5, Mesp1Cre-Tbx1cKO E9.5, Tbx1 Ctrl E9.5 and Tbx1 cKO E9.5. Downstream analysis of scRNA-seq and DEG analysis of the mouse dataset was performed with R-package Seurat3.0[61], as described above. Cluster identity was determined with the marker gene expression pattern reported[35]. Cross-species comparison and classification score assignment were carried out with R-package SingleCellNet[36] standard workflow (https://github.com/pcahan1/singleCellNet).

**Statistics and reproducibility**. All functional measurements included at least four batches of differentiation from each line. For hCAS electrophysiological parameters, dotplots and bar charts were generated from GraphPad Prism. In dotplots, data were presented as mean ± SD. The normality of data (APD and CaT) was assessed by the normality test in GraphPad Prism. For data with normal distribution, statistical significance was first evaluated by Ordinary one-way ANOVA followed by Tukey's multiple comparisons test. For data that were not normally distributed, statistical significance was first evaluated by the Kruskal–Wallis test, followed by Dunn's multiple comparison test. For comparison of the incidence of re-entry events, fisher's exact test was performed with the absolute number of the incidence. A P-value <0.05 after correction for multiple comparisons was considered significant.

**Reporting summary**. Further information on research design is available in the Nature Portfolio Reporting Summary linked to this article.

## Data availability

Single-cell RNA-seq (scRNA-seq) data is available via GEO (GSE186293). Whole genome sequencing data is available at the European Genome-Phenome Archive (EGAS00001006035). Original tracings of electrophysiological assessments in Fig. 4 can be found in Supplementary Data 3.

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

## Acknowledgements

We thank the Genomics & Bioinformatics Cores for providing assistance on scRNA-seq and, Imaging and Flow Cytometry Core of the Centre for PanorOmic Sciences at the University of Hong Kong for flow cytometry and imaging support. We are also grateful for the support of the Ming Wai Lau Centre of Reparative Medicine of Karolinska Institutet via the MWLC Association Member Programme.

## Author contributions

C.H.C., R.A.L., W.K. and Y.F.C. conceived the study. C.H.C., G.L. and N.W. performed hiPSC culture and validation. A.Z., V.A. and J.Z. contributed to variation and structural analyses of the hiPSC. C.H.C., Y.Y.L. and G.L. performed directed cardiac differentiation and flow cytometry validation. C.H.C. and Y.Y.L. performed immunomagnetic cell isolation and prepared cells for scRNA-seq. C.H.C. performed computational analysis for scRNA-seq, hCAS fabrication, measurement and analysis. C.H.C., F.L., W.K. and Y.F.C. contributed to data interpretation. C.H.C., W.K. and Y.F.C. wrote the manuscript.

## Competing interests

The authors declare no competing interests.

## Ethics

Experiments with donated blood samples were approved by the Institutional Review Board of the University of Hong Kong (UW19–506) and proceeded with informed written consent obtained from all subjects.
