## [Peer Review File · Communications Biology]

Reviewers' comments:

Reviewer #1 (Remarks to the Author):

This manuscript reports a number of interesting observations concerning the response of two 22q11.2DS hiPSC cell lines when subjected to a cardiac differentiation protocol. The authors use current techniques for data collection and analysis, but the protocol has not yet been peer-reviewed, although they provide a pdf of a submitted manuscript describing it. I am not in a position to review the related manuscript.

Some of the results obtained are consistent with mouse data using Tbx1 mutants, but the authors have not performed a formal, systematic comparison of data sets. A more extensive comparison should be done, I understand that this may be challenging being mouse and human, but it is not impossible.

The major weakness of this paper is that they have used only 2 cell hiPSC lines from two patients (with some assays they have used only one lines). These lines are notoriously variable in their response to differentiation protocols, beside obviously having different genetic backgrounds from the donor lines. At least some major results should be validated in additional lines. Conclusions drawn from one or two hiPSC lines cannot be considered reliable. In addition, it seems (although not specified) that the single cell transcriptome was evaluated only once, there were no biological replicates.

Aplnr, used in this protocol as a marker for sorting cells, has been reported as a target of Tbx1 (Nomura et al., cited in the manuscript). This would introduce a bias in the experimental approach followed here because potentially the cell population sorted from donor lines may be different from the population sorted from patients' lines. This would be an additional reason to validate results using alternative protocols.

The study is purely descriptive, and results (in particular the transcriptional ones) have not been validated in any way with any other method. Specifically, it would be critical to use an alternative, established differentiation protocol for cardiac differentiation to validate at least some of the results. At the end of this work we have potentially useful observations to be validated and confirmed, but we do not have any additional insights as to the mechanisms by which mutant cells respond differently to differentiation induced by the protocol.

The clinical phenotype and the extent of the deletion of the two patients should be explicitly described (Fig. S1 does not have a sufficient definition).

Reviewer #2 (Remarks to the Author):

1. Overall, the manuscript is well-articulated to address the "Abnormal developmental trajectory and vulnerability to cardiac arrhythmias in tetralogy of Fallot with DiGeorge syndrome".
2. The objective of the study indeed seems novel adding more information and knowledge to the existing research from the same groups, and would definitely be of great interest to translation research communities in pediatric cardiovascular research, where there is a great need for such studies and reports for a proper genetic counseling and prenatal diagnosis.
3. Below are some technical and more subject-related comments and inputs:
 - a. Inclusion of keywords for the abstract would be more informative.
 - b. Full form for the abbreviation might be required for "TOF-ND" and "CPs" in the abstract section.
 - c. Were the differentiated cardiomyocytes been characterized and validated for the cardiomyocyte progenitor or matured markers (hiPSC-CMs) and hiPSC-derived cardiac progenitors (hiPSC-CPs)

quantitatively or qualitatively?

d. Since the cited reference for the cardiac differentiation protocol is still under review, it would be better to mention what day the CMs were FACS sorted (APLNR +), would be better if the sorting (FACS) validation figure with the % of different cell types sorted, could be included under the supplementary section

e. Data for the confirmation and validation of the mesoderm markers (like CD13, Cd56, KDR) or cardiac mesoderm markers (PGDFR- α , Flk-1) would be good

f. Inclusion of page number would be better

g. The reason behind using a human cardiac anisotropic sheet (HCAS) for the study could be elaborated more.

h. Protein validation by western blotting (of at least 2 – 3 significant genes) as future prospect (cardiac related): time-matched comparison, Sc-RNA-seq to be considered

i. The validation for using hCAS for CMs for maturation and functional assessment albeit its drawbacks (limited shelf life) could be reconsidered in the future and reports from a few recent studies have in fact demonstrated efficient maturation via metabolic-pathway modulations, especially through the peroxisome-proliferator-associated receptor (PPAR) and fatty acid (FAO) oxidation.

j. It would be interesting and as a future prospective study, the authors could include or consider the transcriptomic (scRNA-seq) analysis at the hiPSC level, in addition to the hiPSC-CMs and hiPSC-CPs.

k. Details and characteristics of the subjects or patients (TOF-DG: 2; TOF-ND:2; and the healthy control) included in the study (like gender, race, time of diagnosis, shunt details, valve diagnosis, etc.,) need to be included

l. Images of the validated iPSCs for the pluripotency markers OCT3/4, SOX2, SSEA4, and TRA-1-81) in hiPSCs and germ layer markers (AFP for endoderm, α -SMA for mesoderm, and TUB-b-III for ectoderm) in teratoma-formation assay could also be included in the supplementary section.

m. As future prospects (as an extension for the study), it would be informative for the authors to include or consider the following:

Since one of the important diagnosis associated with the tetralogy of Fallot with DiGeorge syndrome is implicated to be congenital heart defects (CHD) that is chiefly linked to the septal defects like ASD, AVSD (NKX2-5) apart from the ventricular defects, the objective of the study to be extended for hiPSC-derived atrial cardiomyocytes (progenitor and mature) as well.

4. The results and conclusions from this study are undeniably original, and convincing and certainly influence further research in the field.

5. The statistical analysis seems appropriate and certainly will be able to reproduce the work, given the level of detail provided.

6. I would highly encourage the transparency and openness of the reviewing process, and would be more than happy to sign the report.

Response to reviewer's comments

Reviewer 1

1. *This manuscript reports a number of interesting observations concerning the response of two 22q11.2DS hiPS cell lines when subjected to a cardiac differentiation protocol. The authors use current techniques for data collection and analysis, but the protocol has not yet been peer-reviewed, although they provide a pdf of a submitted manuscript describing it. I am not in a position to review the related manuscript.*

Thank you for your interest in our observations. The protocol has been submitted to Heliyon for peer review, currently in revision, and bioRxiv as a preprint. We have cited the preprint version in this revised manuscript. In essence, our APLNR sorting protocol is the same as the traditional embryoid body cardiac differentiation, with the same dosage and time of WNT modulation, with the APLNR⁺ cardiac progenitor isolated being a subpopulation of the traditional protocol.

2. *Some of the results obtained are consistent with mouse data using Tbx1 mutants, but the authors have not performed a formal, systematic comparison of data sets. A more extensive comparison should be done, I understand that this may be challenging being mouse and human, but it is not impossible.*

We thank the suggestion of the reviewer. We have further analyzed the mouse dataset and made cross-species comparison between the mouse dataset and our progenitor data. Our D7/8 hiPSC-CPs and D11 cardiomyocytes were indeed found to be similar to the anterior SHF (aSHF) and cardiomyocytes in the mouse dataset. Ectopic gene expression was found exclusively in the multilineage progenitor in the mouse dataset and was absent in the aSHF and cardiomyocytes. These data have been added in the revised manuscript (lines 229 to 242).

We also added discussion on finding in a previous study (reference 39) that demonstrated the trans-regulatory effect of 22q11.2 deletion in primary cell lines derived from the patients with the deletion (lines 332-341). The authors identified dysregulated gene expression along with histone modification changes in the cell lines with 22q11.2 deletion and the histone modification changes were found across the entire genome, including but not limited to chromosome 22q. Although the primary cell lines used were not of cardiac lineage as highlighted in the current manuscript, the global epigenetic change is likely applicable to other cell lines harboring 22q11.2 deletion. Such global epigenomic changes may account for the difference observed in the Tbx1-cko mouse model and our hiPSC-TOF-DG lines.

3. *The major weakness of this paper is that they have used only 2 cell hiPSc lines from two patients (with some assays they have used only one lines). These lines are notoriously variable in their response to differentiation protocols, beside obviously having different genetic backgrounds from the donor lines. At least some major results should be validated in additional lines. Conclusions drawn from one or two hiPS lines cannot be considered*

reliable. In addition, it seems (although not specified) that the single cell transcriptome was evaluated only once, there were no biological replicates.

We have added additional single cell samples (hiPSC-CPs at Day7/8/11) from TOF-DG2 to the analysis and in total there are samples from two control and two TOF-DG lines for all the single cell transcriptome experiments. The newly added dataset confirmed the previous findings of bifurcation of TOF-DG-hiPSC during cardiac differentiation as early as Day 7 and that the bifurcated subgroup is marked by *RGS13* expression (lines 223 to 228). However, the downregulation of cardiac genes found in *RGS13*⁺ hiPSC-CMs (hCAS) was found in the *RGS13*⁺ hiPSC-CPs from TOF-DG1 only. Hence, we have removed the claim of the downregulated cardiac gene expression in the hiPSC-CP scRNA-seq result as well.

While variations in the response to cardiac differentiation from different hiPSC lines and hence the cTNT% among hiPSC lines occur in not only patient but also control-derived cell lines, using the *APLNR*⁺ sorting protocol, we have isolated the hiPSC-CPs and limited the analysis on the cardiac lineage with minimization of influence from the other lineages. Hence, the differentiation is relatively consistent and stable. From the hiPSC-CP and hiPSC-CM scRNA-seq data, our cells are exclusively of cardiac lineage. Although there are variations between the two TOF-DG hiPSC lines in terms of the percentage of cells (hiPSC-CPs and hiPSC-CMs) expressing ectopic genes, these variations should not invalidate our novel findings of the ectopic neural gene expression in TOF-DG-hiPSC-CM and the bifurcation of TOF-DG-hiPSC-CP marked by *RGS13* expression

ScRNA-seq was evaluated once. It is worth noting that many of the iPSC studies published to date have also relied on single evaluation of scRNA-seq, including those published in this esteemed Journal^{1,2}. While we understand the concern over data reproducibility and batch effect, for our study, the scRNA-seq has been carried out for progenitors (D7-11) and cardiomyocytes (D22) from different batches of differentiation and we identified consistently ectopic gene *RGS13* being expressed in a subset of cardiac lineage from TOF-DG-hiPSC only.

Ref 1. Feng, W. *et al.* Computational profiling of hiPSC-derived heart organoids reveals chamber defects associated with *NKX2-5* deficiency. *Commun Biol* **5**, 399, doi:10.1038/s42003-022-03346-4 (2022).

Ref 2. Novak, G. *et al.* Single-cell transcriptomics of human iPSC differentiation dynamics reveal a core molecular network of Parkinson's disease. *Commun Biol* **5**, 49, doi:10.1038/s42003-021-02973-7 (2022).

4. *Aplnr*, used in this protocol as a marker for sorting cells, has been reported as a target of *Tbx1* (Nomura *et al.*, cited in the manuscript). This would introduce a bias in the experimental approach followed here because potentially the cell population sorted from donor lines may be different from the population sorted from patients' lines. This would be an additional reason to validate results using alternative protocols.

From our previous data in the APLNR manuscript³, we found *APLNR* expression in very early mesendoderm progenitor (Day4 *in vitro* cardiac differentiation) in which the *TBX1* expression was minimal (In text Fig 1). It is therefore unlikely that APLNR is downstream of *TBX1*.

In text Fig 1. Progenitor gene expression on Day4 *in vitro* cardiac differentiation

Although we could not validate the relationship between APLNR and TBX1 from the mouse data, the scRNA-seq data on Day 7 progenitors in the current manuscript was found to have comparable TBX1 expression level between the control and TOF-DG lines(Fig 3C-D). Therefore, the differentiation protocol used is not biased to the TBX1 expression.

Ref 3. Lam, Y.-Y. *et al.* APLNR marks a cardiac progenitor derived with human induced pluripotent stem cells. *bioRxiv*, 2023.2002.2022.529606, doi:10.1101/2023.02.22.529606 (2023).

5. *The study is purely descriptive, and results (in particular the transcriptional ones) have not been validated in any way with any other method. Specifically, it would be critical to use an alternative, established differentiation protocol for cardiac differentiation to*

validate at least some of the results. At the end of this work we have potentially useful observations to be validated and confirmed, but we do not have any additional insights as to the mechanisms by which mutant cells respond differently to differentiation induced by the protocol.

While the present piece of work may be regarded as a descriptive study, this represents the first attempt to understand the intricate transcriptomic changes along the cardiac differentiation, from cardiac progenitors to terminally differentiated cardiomyocytes, in the most common cyanotic congenital heart disease (tetralogy of Fallot) in the context of the deletion syndrome encountered in the clinical setting (DiGeorge syndrome). Hence, the findings from this study would likely pave the way for further mechanistic studies (e.g. exploration of the significance of TBX1 in the ectopic gene expression in DG, exploration of potential chromatin changes induced by 22q22.1 deletion) (lines 357-362).

To increase the robustness of our data, as alluded to in point 3, we have added hiPSC-CP-scRNA-seq data to support the *RGS13* expression finding. Analytic tools with high resolution, like scRNA-seq, is required to assess the heterogeneity among the differentiated cells. Traditional proteomic approach might not be enough to validate the transcriptomic finding.

6. *The clinical phenotype and the extent of the deletion of the two patients should be explicitly described (Fig. S1 does not have a sufficient definition).*

The clinical phenotype has been added as supplementary table 1. Fig. S3 has been updated to show the extent of the deletion.

Reviewer 2

- 1) *Overall, the manuscript is well-articulated to address the “Abnormal developmental trajectory and vulnerability to cardiac arrhythmias in tetralogy of Fallot with DiGeorge syndrome”. The objective of the study indeed seems novel adding more information and knowledge to the existing research from the same groups, and would definitely be of great interest to translation research communities in pediatric cardiovascular research, where there is a great need for such studies and reports for a proper genetic counseling and prenatal diagnosis.*

We thank the kind comments of the reviewer.

- 2) *Below are some technical and more subject-related comments and inputs:*
 - a) *Inclusion of keywords for the abstract would be more informative.*
The inclusion of keywords is not a standard format for *Communication Biology*.
 - b) *Full form for the abbreviation might be required for “TOF-ND” and “CPs” in the abstract section.*
The full form has been provided (line 48-49).

- c) *Were the differentiated cardiomyocytes been characterized and validated for the cardiomyocyte progenitor or matured markers (hiPSC-CMs) and hiPSC-derived cardiac progenitors (hiPSC-CPs) quantitatively or qualitatively?*

The differentiated cardiomyocytes were validated qualitatively with the matured markers cardiac troponin II (TNNT2/cTNT), only hiPSC-CMs with over 70% cTNT were used for CAS experiments (line 479).

- d) *Since the cited reference for the cardiac differentiation protocol is still under review, it would be better to mention what day the CMs were FACS sorted (APLNR +), would be better if the sorting (FACS) validation figure with the % of different cell types sorted, could be included under the supplementary section*

Thank you for the suggestion. We have added a supplementary figure (Fig S6) with the schematics of the differentiation protocol. As the Day5 cardiac progenitors were sorted out by MS columns and OctoMACS™ Separator instead of FACS flow cytometry (line 434-459), we could not provide a FACS validation figure.

- e) *Data for the confirmation and validation of the mesoderm markers (like CD13, Cd56, KDR) or cardiac mesoderm markers (PGDFR- α , Flk-1) would be good*
Validation with cardiac mesoderm markers (KDR/PDGFR- α) has been performed in the previous APLNR manuscript as described above. The majority of APLNR sorted progenitors are positive for the KDR/PDGFR- α . (In text Fig 2)

In text Fig 2. KDR and PDGFRA expression in APLNR⁺ progenitor, data extracted from the protocol manuscript

- f) *Inclusion of page number would be better*

Page numbers have now been added.

- g) *The reason behind using a human cardiac anisotropic sheet (HCAS) for the study could be elaborated more.*

The reason for using hCAS is for the measurement of various electrophysiological parameters with this syncytium of cardiomyocytes (lines 96-98). It allows the

detection and recording of re-entrant arrhythmia, an important clinical feature observed in TOF-DG2-hCAS.

- h) *Protein validation by western blotting (of at least 2 – 3 significant genes) as future prospect (cardiac related): time-matched comparison, Sc-RNA-seq to be considered*

While protein validation is a possible consideration, the heterogeneity in the hiPSC-CMs may pose uncertainty in data interpretation. As shown by scRNA-seq, only a subset of hiPSC-CMs from TOF-DG expressed ectopic genes and showed downregulation of cardiac gene expression. Protein assays with higher cellular resolution would be required.

- i) *The validation for using hCAS for CMs for maturation and functional assessment albeit its drawbacks (limited shelf life) could be reconsidered in the future and reports from a few recent studies have in fact demonstrated efficient maturation via metabolic-pathway modulations, especially through the peroxisome-proliferator-associated receptor (PPAR) and fatty acid (FAO) oxidation.*

Thank you for this interesting suggestion and we agree that further works along this line should be performed.

- j) *It would be interesting and as a future prospective study, the authors could include or consider the transcriptomic (scRNA-seq) analysis at the hiPSC level, in addition to the hiPSC-CMs and hiPSC-CPs.*

Thank you for the suggestion. Inclusion of the hiPSC scRNA-seq would be valuable in evaluating the developmental trajectory from hiPSC to hiPSC-CPs and -CMs in the future.

- k) *Details and characteristics of the subjects or patients (TOF-DG: 2; TOF-ND:2; and the healthy control) included in the study (like gender, race, time of diagnosis, shunt details, valve diagnosis, etc.,) need to be included.*

The clinical phenotype has been added as **Supplementary table 1**.

- l) *Images of the validated iPSCs for the pluripotency markers OCT3/4, SOX2, SSEA4, and TRA-1-81) in hiPSCs and germ layer markers (AFP for endoderm, α -SMA for mesoderm, and TUB-b-III for ectoderm) in teratoma-formation assay could also be included in the supplementary section.*

Supplementary figures with the immunohistochemistry of the pluripotency markers in hiPSCs (**Fig S1**) and germ layer markers in embryoid formation assay (**Fig S2**) have been added.

- m) *As future prospects (as an extension for the study), it would be informative for the authors to include or consider the following:*

Since one of the important diagnosis associated with the tetralogy of Fallot with DiGeorge syndrome is implicated to be congenital heart defects (CHD) that is chiefly linked to the septal defects like ASD, AVSD (NKX2-5) apart from the ventricular defects, the objective of the study to be extended for hiPSC-derived atrial cardiomyocytes (progenitor and mature) as well.

Thank you for the suggestion. It would be a great idea to compare the atrial and ventricular DG-hiPSC-CMs and see if the finding in ventricular DG-hiPSC-CMs holds true in the atrial counterparts. Unfortunately, the hiPSC-CMs derived from the current protocol biases towards ventricular expression (as indicated by the MYH7 and MYL2 expression in the hCAS scRNA-seq result). We would need to further develop our protocol to generate atrial hiPSC-CMs from the same hiPSC-CPs.

- 3) *The results and conclusions from this study are undeniably original, and convincing and certainly influence further research in the field. The statistical analysis seems appropriate and certainly will be able to reproduce the work, given the level of detail provided.*

We thank the reviewer for the kind comment.

Reviewers' comments:

Reviewer #1 (Remarks to the Author):

I believe that the authors have addressed the reviewers' comments in a satisfactory manner. I am requesting some minor adjustments to the text and addition of some data to the supplementary material.

Abstract:

- "upregulated neural gene expression", also elsewhere in the text. This expression is excessively vague, it requires a better description, for example how many genes have been considered, how many of these are significantly upregulated in deleted samples?

- the sentence "Transcriptomic profiling of the in vitro cardiac progenitors revealed early bifurcation in the trajectory of TOF-DG-hiPSC cardiac differentiation" is obscure at this point of the manuscript, authors should make an effort to explain, in an efficient and brief manner, what do they mean by "early bifurcation".

- The sentence " These findings were found only in the TOF-DG but not TOF-with no DG -(ND)" should be presented with a caveat because the number of observations is so small that differences have no statistical robustness.

- Refs 27 and 28 are correct but there are other earlier references that the authors should also cite, for example Lindsay et al, Nature 1999, the first model of DG.

- Fig. 2D GO term genes. Genes included in the various terms and found in the dataset should be indicated, perhaps in a worksheet within the supplementary material.

Typos, etc.

Line 215 "micodeletion";

line 386 double word;

Please check thoroughly...

Reviewer #2 (Remarks to the Author):

Thank you for the response to my comments and suggestions.

However, regarding few of the comments that I had raised earlier has not been specifically addressed or included in the manuscript as follows:

1. The comment regarding the validation/characterization for either progenitor or mature cardio myocyte markers:

a) Were the differentiated cardiomyocytes been characterized and validated for the cardiomyocyte progenitor or matured markers (hiPSC-CMs) and hiPSC-derived cardiac progenitors (hiPSC-CPs) quantitatively or qualitatively?

The differentiated cardiomyocytes were validated qualitatively with the matured markers cardiac troponin II (TNNT2/cTNT), only hiPSC-CMs with over 70% cTNT were used for CAS experiments (line 479).

I do not see either the qualitative or quantitative images as proof for this validation, except for the "line 516" mentioning on "day 12 hiPSC-CMs (with cTNT % > 70%)"

2. Regarding the validation for the sorted APLNR + cardio myocytes, I understand that the authors have used MACS sorter, and have added just the schematic figure (fig S6) of the overall protocol. The MACS sorted APLNR + cardio myocytes however, could be quantitatively analyzed after MACS sorting in flow cytometry analyzer (like Celesta, Fortessa, Cytex Aurora, and so on) to give the exact % of the cell population. I hope this comment would make sense.

3. The validation for the cardiac mesoderm markers (KDR/PGDFR- α) quantification that has been performed in the previous APLNR manuscript as mentioned by the authors, I am wondering if the represented data shown in the rebuttal section (e), the gene expression data?

Reviewer #1

I am requesting some minor adjustments to the text and addition of some data to the supplementary material.

Abstract:

-“upregulated neural gene expression”, also elsewhere in the text. This expression is excessively vague, it requires a better description, for example how many genes have been considered, how many of these are significantly upregulated in deleted samples?

The ectopic neural gene was expressed in cluster B0 and the number of DEGs varied when compared to other clusters (B1/B2/B3). We have added the number of DEGs identified and the genes considered in the GO-term in the text (lines 184-185) and in figures 2D and 4F. We have further included an Excel file (as supplementary material 1 that includes all the DEG and GO analysis results.

-the sentence “Transcriptomic profiling of the in vitro cardiac progenitors revealed early bifurcation in the trajectory of TOF-DG-hiPSC cardiac differentiation” is obscure at this point of the manuscript, authors should make an effort to explain, in an efficient and brief manner, what do they mean by “early bifurcation”.

We added “as marked by RGS13 expression” to the abstract (line 46) to explain the bifurcation.

-Refs 27 and 28 are correct but there are other earlier references that the authors should also cite, for example Lindsay et al, Nature 1999, the first model of DG.

Thank you for the suggestion, which has been added in the revision (ref 28).

-Fig. 2D GO term genes. Genes included in the various terms and found in the dataset should be indicated, perhaps in a worksheet within the supplementary material.

An Excel file containing the DEG and GO term analysis of cluster B0 has been included as supplementary material 1.

Typos, etc.

Line 215 “micodeletion”

Line 386 double word;

Please check thoroughly...

We have checked and made changes accordingly.

Reviewer #2

Thank you for the response to my comments and suggestions.

However, regarding few of the comments that I had raised earlier has not been specifically addressed or included in the manuscript as follows:

1. The comment regarding the validation/characterization for either progenitor or mature cardiomyocyte markers:

- a. Were the differentiated cardiomyocytes been characterized and validated for the cardiomyocyte progenitor or matured markers (hiPSC-CMs) and hiPSC-derived cardiac progenitors (hiPSC-CPs) quantitatively or qualitatively?

The differentiated cardiomyocytes were validated qualitatively with the matured markers cardiac troponin II (TNNT2/cTNT), only hiPSC-CMs with over 70% cTNT were used for CAS experiments.

I do not see either the qualitative or quantitative images as proof for this validation, except for the “line 516” mentioning on “day12 hiPSC-CMs (with cTNT %>70%)”

To address the concern over the validation for the cardiomyocytes, we have included the cTNT% of cardiomyocytes from each cell line (Fig. S6C) and representative flow-cytometry images of the cTNT expression analysis (Fig. S6D).

2. Regarding the validation for the sorted APLNR+ cardiomyocytes, I understand that the authors have used MACS sorter, and have added just the schematic figure (fig S6) of the overall protocol. The MACS sorted APLNR+ cardiomyocytes however, could be quantitatively analyzed after MACS sorting in flow cytometry analyzer (like Celesta, Fortessa, Cytex Aurora, and so on) to give the exact % of the cell population. I hope this comment would make sense.

Although analysis of the proportion of MACS sorted APLNR+ progenitor was not performed in our routine differentiation, we did keep track of the input cell number and the number of sorted APLNR+ progenitors. The percentage of APLNR+ progenitors from each cell line in 10 differentiation has been included (Fig. S6B).

3. The validation for the cardiac mesoderm markers (KDR/PDGFR-a) quantification that has been performed in the previous APLNR manuscript as mentioned by the authors, I am wondering if the represented data shown in the rebuttal section (e), the gene expression data?

Our apology for causing the confusion. The data represent the proportion of cells that expressed the corresponding protein, as analyzed by flow-cytometry. For details, please refer to our manuscript that has been just published online (<https://doi.org/10.1016/j.heliyon.2023.e18243>). This reference is also cited in the revised text (ref 32).

REVIEWERS' COMMENTS:

Reviewer #2 (Remarks to the Author):

Thank you for responding to my comments and suggestions.
However, please address the following:

1. Regarding my query about the characterization and validation of the hiPSC-derived progenitor or matured cardio myocyte markers (hiPSC-CMs) quantitatively or qualitatively, to which the authors have responded as below:

"validated qualitatively with the matured markers cardiac troponin II (TNNT2/cTNT), only hiPSC-CMs with over 70% cTNT were used for CAS experiments"

If that is the case, it would be better if they show the qualitatively validated immunofluorescence images for the mature cardiac markers for cardiac troponin II (TNNT2/cTNT).

This would be more convincing for this particular query.

2. In addition, the authors have stated that they have included the cTNT% of cardio myocytes from each cell line (Fig. S6C) and representative flow-cytometry images of the cTNT expression analysis (Fig.S6D).

These included images S6C and S6D, looks too vague and the scatter plot looks too unclear and especially the FACS data S6D is not convincing. Would it possible for the authors to enclose the raw data (flow cytometry data) to support this query?

3. Similarly, for my query regarding the validation for the (MACS) sorted APLNR+ cardio myocytes, for which the authors have responded that

"They did keep track of the input cell number and the number of sorted APLNR+ progenitors. The percentage of APLNR+ progenitors from each cell line in 10 differentiation has been included (Fig. S6B)"

It would be better if the authors demonstrate their data (even with the input cell number) before and after MACS sorting, to show exactly the % increase in the APLNR+ cardio myocytes after MACS sorting, to give a rough sense of the % of the non-cardio myocyte population before and after. Therefore, the figure S6B can be modified accordingly if possible.

Reviewer #2 (Remarks to the Author):

Thank you for responding to my comments and suggestions. However, please address the following:

1. Regarding my query about the characterization and validation of the hiPSC-derived progenitor or matured cardio myocyte markers (hiPSC-CMs) quantitatively or qualitatively, to which the authors have responded as below:

“validated qualitatively with the matured markers cardiac troponin II (TNNT2/cTNT), only hiPSC-CMs with over 70% cTNT were used for CAS experiments”

If that is the case, it would be better if they show the qualitatively validated immunofluorescence images for the mature cardiac markers for cardiac troponin II (TNNT2/cTNT).

This would be more convincing for this particular query.

We apologize for the typo and the confusion. It should be validated ‘quantitatively’ instead of ‘qualitatively’ as the cardiac troponin II expression was evaluated with flow cytometry.

2. In addition, the authors have stated that they have included the cTNT% of cardio myocytes from each cell line (Fig. S6C) and representative flow-cytometry images of the cTNT expression analysis (Fig.S6D).

These included images S6C and S6D, looks too vague and the scatter plot looks too unclear and especially the FACS data S6D is not convincing. Would it possible for the authors to enclose the raw data (flow cytometry data) to support this query?

We have adjusted the transparency of the data point in the scatter plot (S6C) to make it more solid and readable. The clarity of S6D has been improved as well. We have also included the raw flow cytometry data of one of our cTNT expression analysis as an example.

3. Similarly, for my query regarding the validation for the (MACS) sorted APLNR+ cardio myocytes, for which the authors have responded that

“They did keep track of the input cell number and the number of sorted APLNR+ progenitors. The percentage of APLNR+ progenitors from each cell line in 10 differentiation has been included (Fig. S6B)”

It would be better if the authors demonstrate their data (even with the input cell number) before and after MACS sorting, to show exactly the % increase in the APLNR+ cardio myocytes after MACS sorting, to give a rough sense of the % of the non-cardio

myocyte population before and after. Therefore, the figure S6B can be modified accordingly if possible.

This point was addressed by the data shown in Fig 4B and 4C in our published protocol (doi: 10.1016/j.heliyon.2023.e18243), which clearly showed the percentage of cardiomyocyte (cTNT⁺) and non-cardiomyocyte (cTNT⁻) derived from unsorted/APLNR⁺/APLNR⁻ progenitors. This article has been cited already in our revised submission.